# Communication between distinct subunit interfaces of the cohesin complex promotes its topological entrapment of DNA

**Vincent Guacci, Fiona Chatterjee, Brett Robison, Douglas E Koshland\***

Department of Molecular and Cell Biology, University of California, Berkeley, Berkeley, United States

**Abstract** Cohesin mediates higher order chromosome structure. Its biological activities require topological entrapment of DNA within a lumen(s) formed by cohesin subunits. The reversible dissociation of cohesin's Smc3p and Mcd1p subunits is postulated to form a regulated gate that allows DNA entry and exit into the lumen. We assessed gate-independent functions of this interface in yeast using a fusion protein that joins Smc3p to Mcd1p. We show that in vivo all the regulators of cohesin promote DNA binding of cohesin by mechanisms independent of opening this gate. Furthermore, we show that this interface has a gate-independent activity essential for cohesin to bind chromosomes. We propose that this interface regulates DNA entrapment by controlling the opening and closing of one or more distal interfaces formed by cohesin subunits, likely by inducing a conformation change in cohesin. Furthermore, cohesin regulators modulate the interface to control both DNA entrapment and cohesin functions after DNA binding.
DOI: https://doi.org/10.7554/eLife.46347.001

**\*For correspondence:**
koshland@berkeley.edu

**Competing interests:** The authors declare that no competing interests exist.

## Introduction

Cohesin is a member of a family of Smc complexes required to higher order chromosome structure and dynamics (*Onn et al., 2008*). Cohesin tethers sister chromatids together from S phase though metaphase to ensure proper segregation at anaphase (*Guacci et al., 1997*; *Michaelis et al., 1997*). Cohesin binds chromosomes at multiple sites, termed CARs, to enable tethering (*Blat and Kleckner, 1999*; *Megee et al., 1999*; *Laloraya et al., 2000*). Cohesin is also important for mitotic chromosome condensation and proper gene expression (*Guacci et al., 1997*; *Rollins et al., 2004*). These functions of cohesin may arise from its ability to partition chromosomes into distinct segments that could be packaged into domains of condensation and transcription (*Guacci et al., 1997*; *Hartman et al., 2000*; *Lavoie et al., 2002*; *Eagen, 2018*). Finally, cohesin is loaded at sites of DNA breaks to facilitate both cohesion and repair (*Ström et al., 2007*; *Unal et al., 2007*). Understanding how cohesin carries out all these biological functions requires elucidating how cohesin binds to DNA, how binding is maintained and how binding is abrogated.

Cohesin must stably bind DNA to execute its different functions. In budding yeast, the four evolutionarily conserved subunits of cohesin are Smc1p, Smc3p, Mcd1p/Scc1p and Scc3p (*Guacci et al., 1997*; *Michaelis et al., 1997*; *Losada et al., 1998*). Biochemical studies indicate that DNA is topologically entrapped within cohesin, and such entrapment can occur within a trimer formed by Smc1p, Smc3p and Mcd1p (*Ivanov and Nasmyth, 2007*; *Haering et al., 2008*; *Gligoris et al., 2014*). The trimer architecture suggests it forms two lumens capable of trapping DNA. A large lumen is formed by dimerization of the Smc1p and Smc3p hinge domains at one end and dimerization of the two head domains at the other (*Figure 1A*). A smaller lumen is formed by the binding of

**Figure 1.** Cohesin structure. (A) Cartoon showing the cohesin complex and Pds5p. The three interfaces that can be chemically crosslinked to trap DNA within the Smc1p-Smc3p-Mcd1p trimer are marked by arrows. These are 1) Smc3p Mcd1p interface, 2) Smc1p hinge Smc3p hinge dimer interface, and 3) Smc1p Mcd1p interface. (B) Crystal structure of the Smc3p coiled-coil and Mcd1p NHD domain interface. Crystal structure entry PDB 4U × 3 of the Smc3 head domain + short coiled-coil (Smc3 CC; blue) and Mcd1/Scc1 NHD (Mcd1 N; red) is shown. Right Side is enlargement showing essential residues in the interface as spheres.

DOI: https://doi.org/10.7554/eLife.46347.002

the Mcd1p C-terminal domain to the Smc1p head domain, and the binding of the Mcd1 N-terminal helical domain (NHD) to coiled coil residues emerging from the Smc3p head domain (*Figure 1B*) (*Haering et al., 2008*; *Gligoris et al., 2014*). While initial models for topological entrapment favored DNA in the large lumen, studies of cohesin and other related SMC complexes have suggested entrapment by the small lumen as well (*Stigler et al., 2016*).

To stably entrap DNA, one or more interfaces of the cohesin trimer must open to allow DNA to enter one of these lumens (*Figure 1A*). Controlling the opening and closing of these interfaces provides a mechanism for regulators to promote cohesin loading onto DNA, stable binding to DNA, and release from DNA. A dimeric Scc2p/Scc4p complex termed the cohesin loader is required for cohesin to bind DNA (*Ciosk et al., 2000*). Evidence indicates that DNA enters through opening of the hinge dimer interface and not through interfaces formed by head dimerization or interfaces between Smc3p and Mcd1p or Mcd1p and Smc1p (*Gruber et al., 2006*). However, in vitro biochemical experiments suggested that Scc2/Scc4 may load cohesin onto DNA by the opening of the interface between Smc3p and Mcd1p (*Murayama and Uhlmann, 2015*). In this context, the Smc3p Mcd1p interface is a potential entry gate for topological entrapment.

Once loaded onto DNA, cohesin can either remain stably bound to it or be released. Toggling between these two states has been postulated to result from the antagonistic activities of the cohesin regulators, Eco1p, Pds5p and Wpl1p acting on or near the Smc3p Mcd1p interface. Eco1p (Ctf7p) mediated acetylation of the Smc3p K112 K113 head domain residues promotes stable cohesin DNA binding and cohesion (*Skibbens et al., 1999*; *Tóth et al., 1999*; *Rolef Ben-Shahar et al., 2008*; *Unal et al., 2008*; *Bloom et al., 2018*). Once cohesion is established, Pds5p promotes cohesion maintenance as well as condensation (*Hartman et al., 2000*; *Stead et al., 2003*; *Noble et al., 2006*; *Robison et al., 2018*). Wpl1p binds to Pds5p to promote cohesin's release from DNA and serves to antagonize both cohesion and condensation (*Gandhi et al., 2006*; *Kueng et al., 2006*; *Rowland et al., 2009*; *Guacci and Koshland, 2012*; *Lopez-Serra et al., 2013*; *Bloom et al., 2018*).

Additional studies of Eco1p, Wpl1p and Pds5p have suggested a molecular basis for their ability to toggle cohesin binding to DNA. The binding of an Mcd1p N-terminal cleavage fragment to Smc3p is destabilized by Wpl1p, but is stabilized by Smc3p acetylation or by mutants that bypass *eco1* function (*Chan et al., 2012*; *Beckouët et al., 2016*). These experiments led to an exit gate model whereby DNA can escape cohesin entrapment by Wpl1p-dependent opening the Smc3p Mcd1p interface. Smc3p acetylation would inhibit Wpl1p, thereby keeping the interface closed and DNA topologically bound (*Chan et al., 2012*; *Beckouët et al., 2016*). Thus, the Smc3p Mcd1p interface has been postulated to serve as a regulated DNA exit gate and as well as a regulated entrance gate.

While compelling in its simplicity, viewing the function of the Smc3p Mcd1p interface and cohesin regulators solely through the lens of a putative DNA gate has been challenged by several observations. First, cohesin is stably bound to DNA when Smc3p K112 K113 acetylation is prevented by mutating the critical lysines (*smc3 K112R K113R*) (*Unal et al., 2008*; *Rowland et al., 2009*). Thus, acetylation of these residues is not needed to stabilize topological entrapment of DNA in vivo. However, both *smc3 K112R K113R* and *eco1Δ wpl1Δ* mutants exhibit a dramatic defect in sister chromatid cohesion (*Rowland et al., 2009*; *Sutani et al., 2009*; *Chan et al., 2012*; *Guacci and Koshland, 2012*). These results suggest that acetylation promotes cohesion by an additional step beyond preventing DNA release through an exit gate. Finally, recent experiments suggest that the function of cohesin and other Smc complexes requires a conformational change of the coiled coil, likely driven by the head ATPase activity (*Soh et al., 2015*). The binding of Mcd1p to both the head domain of Smc1 and the coiled coil of Smc3 provides a potential mechanism to transduce the ATP state of the head domain to initiate a conformation change of the coiled coils.

Here, we use a fusion protein of Smc3p and Mcd1p to permanently shut the putative DNA gate as a tool to evaluate gate-independent functions of this interface. We examined whether the fusion could suppress the need for the cohesin loader Scc2p, the Scc3p cohesin subunit, Pds5p and Smc3p K112 K113 acetylation. Our results show that in vivo all these regulators of cohesin promote DNA binding or cohesion by mechanisms independent of opening the Smc3p Mcd1p interface. Furthermore, mutations altering the interface reveal it has a gate-independent activity essential for cohesin to bind DNA in vivo. These observations suggest new models for the Smc3p Mcd1p interface in topological entrapment and as a target of cohesin regulators.

# Results

We wanted to assess whether cohesin regulators control cohesin function by modulating the transient opening of the Smc3p Mcd1p interface. To do so, we used a previously characterized gene fusion in which the open-reading frame (ORF) of *MCD1* was placed in frame at the end of the ORF for *SMC3* (*Gruber et al., 2006*). The product of this gene fuses the Mcd1p N-terminus to the Smc3p C-terminus so this putative DNA gate cannot open. It supports viability as sole source of both Smc3p and Mcd1p. Consequently, regulators that act solely by stabilizing the exit gate should no longer be needed for cohesin function in fusion bearing strains.

We used the Smc3p Mcd1p fusion to test gate-dependent functions of three proteins associated with the trimer cohesin, Scc2p and Pds5p regulators as well as the the Scc3p cohesin subunit. Scc3p is required for cohesin binding to DNA, cohesion and condensation (*Tóth et al., 1999*; *Roig et al., 2014*; *Orgil et al., 2015*). Since Scc2p, Scc3p and Pds5p are subunits of complexes that bind near the interface (*Rowland et al., 2009*; *Orgil et al., 2015*), they are in a position to modulate the Smc3p Mcd1p interface to control its dissociation to allow entry or exit of DNA.

To assess the importance of the dissociation of Smc3 Mcd1 interface for these three regulatory proteins, we first built haploid strains bearing the fusion gene as the sole source of both Smc3p and Mcd1p function. We then replaced the endogenous *SCC2, SCC3*, *PDS5* genes with conditional alleles containing a C-terminal 3V5 tag and an auxin-inducible degron (AID) called *SCC3-AID*, *SCC2-AID* or *PDS5-AID*. The AID degron enabled the rapid degradation of each of these proteins upon auxin addition (*Eng et al., 2014*). As a control, we placed the same *AID* constructs in an otherwise wild-type haploid strain to compare the effects of their depletion with and without the fusion. For clarity, strains that express the normal configuration of Smc3p and Mcd1p as separate subunits are referred to as normal strains or as having normal cohesin. Strains that have the Smc3p-Mcd1p fusion are referred to as fusion strains or as having fusion cohesin.

## Fusion strains still require Scc2p, Scc3p and Pds5p for viability and sister chromatid cohesion

We first assessed whether the fusion bypasses the need for Scc2p, Scc3p and Pds5p for cell viability. Normal and fusion strains with or without AID tagged proteins were grown to saturation and then plated in 10-fold serial dilution either in the presence or absence of auxin. As expected, normal strains depleted for Scc2p-AID, Scc3p-AID or Pds5p-AID were inviable (*Figure 2A and B* top panel) (*Eng et al., 2014*). Depletion of these same proteins in fusion strains was also lethal (*Figure 2A and B* bottom panels). The inviability of fusion strains indicates that Scc2p, Scc3p and Pds5p are required for an essential cohesin biochemical activity other than blocking exit through the putative Smc3p Mcd1p gate.

To begin to understand the exit gate-independent molecular functions of Scc2p, Scc3p and Pds5p, we assayed sister chromatid cohesion in our AID strains with and without the fusion. We used a regimen of synchronous arrest in mid-M phase as follows. Cultures were arrested in G1 where the AID-tagged proteins depleted. Cells were released from G1 into media containing auxin and nocodazole to re-arrest cells in mid-M phase under AID depletion conditions (Materials and methods and *Figure 3—figure supplement 1A*). Cell cycle arrest and depletion of AID-tagged proteins were confirmed by FACS (*Figure 3—figure supplements 1B* and *2A*) and western blot (*Figure 3—figure supplements 1C,D* and *2B*). Mid-M arrested cells were assayed for cohesion at a *CEN*-distal (*LYS4*) locus on chromosome IV using the GFP-LacI and LacO system (Materials and methods). In budding yeast, after replication the sister chromatids are so closely associated they cannot be resolved as individual chromatids by fluorescence microscopy (*Guacci et al., 1994*). Therefore, in the LacO-LacI assay, haploids in mid-M exhibit a single GFP focus, whereas cells that are defective for cohesion have two GFP spots, one from each sister chromatid.

As expected, mid-M arrested cells with normal cohesin had cohesion so two GFP spots were detected in only 5–10% of cells. Whereas, cells depleted of Scc2p-AID, Scc3p-AID, or Pds5p-AID had a major cohesion defect as evidenced by more than 75% of cells having two GFP spots, consistent with previous results (*Figure 3A–B*, top panels) (*Eng et al., 2014*). Most mid-M arrested cells with fusion cohesin had cohesion, although a modest cohesion defect was observed (~25% cells with two GFP spots) similar to that seen in previous studies (*Figure 3A–B*, bottom panels) (*Gruber et al., 2006*; *Bloom et al., 2018*). In contrast, fusion cells depleted for Scc2p, Scc3p or Pds5p had severe

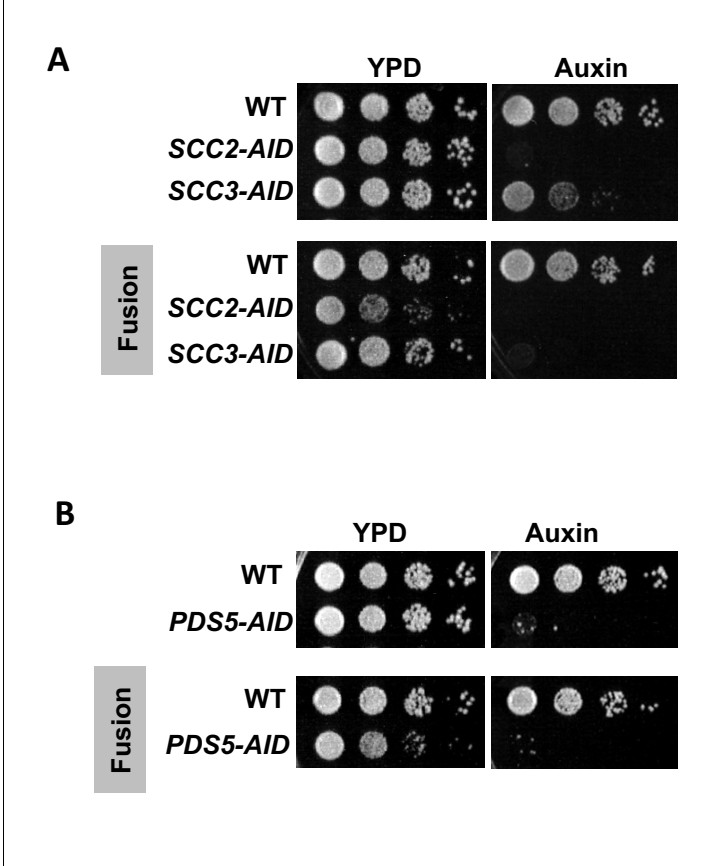

**Figure 2.** Cohesin regulators are required fusion cohesin function. (A-B) Haploid strains with normal cohesin or fusion were grown to saturation, then plated at 10-fold serial dilution onto YPD alone or containing auxin (750 μM) and incubated at 23°C for 3 days. (A) Scc2p-AID and Scc3p-AID depletion causes in inviability in normal and fusion cohesion strains. Top panel shows strains with normal cohesin, WT (VG3620-4C), containing *SCC2-3V5-AID* (VG3630-7A) or *SCC3-3V5-AID* (VG3808-1A). Bottom panel shows strains with fusion cohesin, WT (VG3940-2D) or with *SCC2-3V5-AID* (VG3945-1A) or *SCC3-3V5-AID* (VG3946-7B). (B) Pds5p-AID depletion causes in inviability in normal and fusion cohesion strains. Top panel shows strains with normal cohesin, WT (VG3620-4C) or containing *PDS5-3V5-AID2* (VG3954-10C). Bottom panel shows strains with fusion cohesin, WT (VG3940-2D) or containing *PDS5-3V5-AID2* (VG3955-4D).

DOI: https://doi.org/10.7554/eLife.46347.003

cohesion defects similar in magnitude to that seen after depletion in cells with normal cohesin (*Figure 3A–B*, bottom panels). These results show that the all three proteins play critical roles in sister chromatid cohesion independent of any putative role in keeping the Smc3p Mcd1p interface closed.

## The Smc3p Mcd1p fusion does not suppress the effects of Scc2p, Scc3p and Pds5p depletion on cohesin binding to DNA

To understand further the molecular basis for the gate-independent functions of cohesin regulators, we compared the effect of Scc2p-AID and Scc3p-AID depletion on normal and fusion cohesin binding to chromosomes in cells subjected to our regimen of synchronous arrest. These cells were processed for chromatin immunoprecipitation (ChIP) using anti-Mcd1p antibodies (Materials and methods). In wild-type cells, fusion cohesin bound with the same pattern as normal cohesin at a pericentric *CAR* (*CARC1*) and an arm *CAR* (*TRM1*) locus as well as immediately adjacent to *CEN4* and *CEN14*. (*Figure 3C*). Thus, the minor defect in cohesion observed in the fusion strain was not due to effects on fusion cohesin levels or distribution on chromosomes. For both normal and fusion cohesin, depletion of Scc2p-AID or Scc3p-AID completely abolished cohesin binding at peri-centric *CARC1*

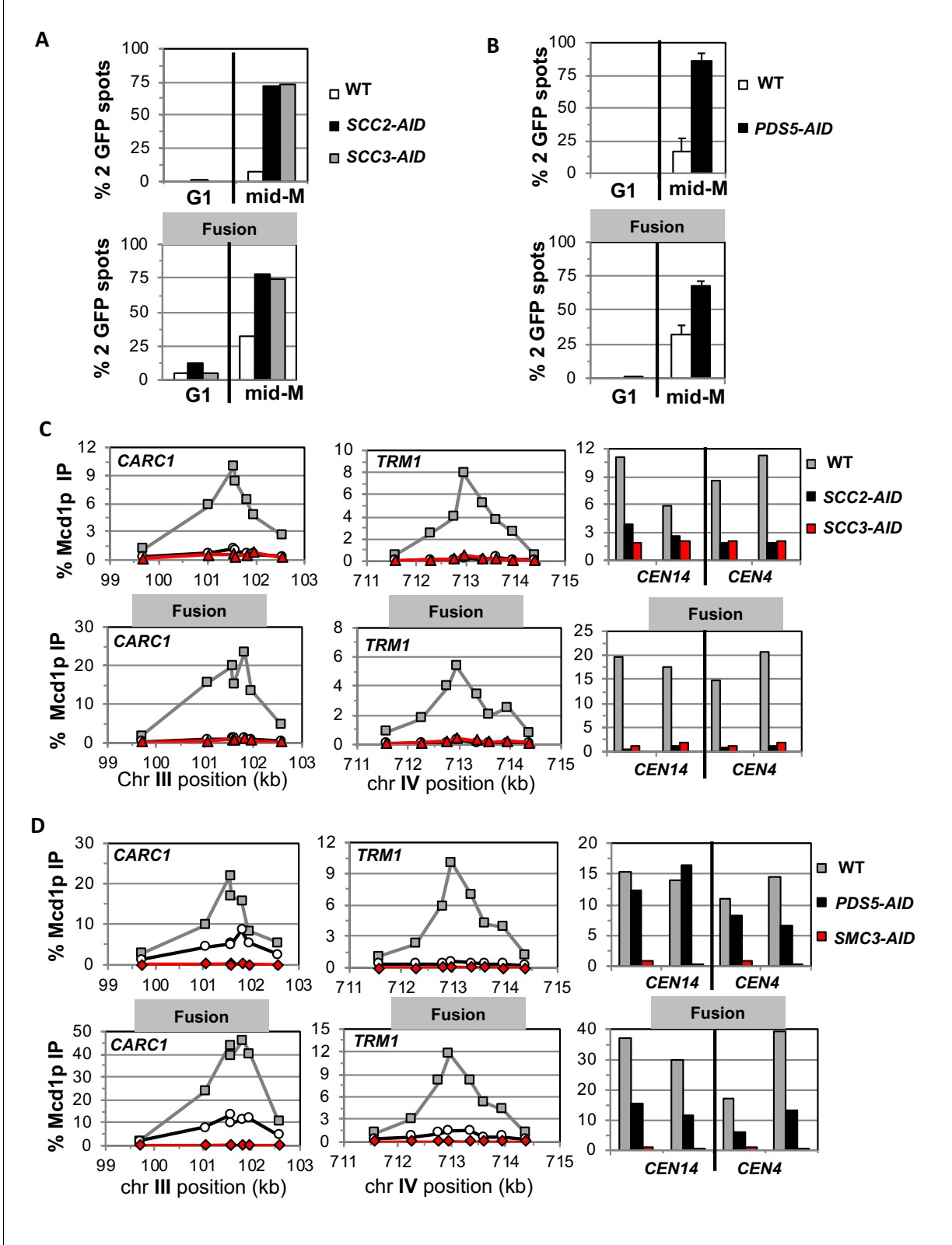

**Figure 3.** Fusion cohesin requires cohesin regulators for sister chromatid cohesion and cohesin binding to DNA. Haploids strains with normal cohesin or fusion cohesin from *Figure 2* were grown to mid-log phase then arrested in G1 using α factor, auxin added to induce loss of AID-tagged proteins then released into media containing nocodazole and auxin to arrest in mid-M phase under AID depletion conditions as described for synchronous mid-M phase arrest in Materials and methods. Cells were fixed and processed to monitor loss of sister chromatid cohesion and for ChIP to monitor

*Figure 3 continued on next page*

*Figure 3 continued*

cohesin DNA binding. (**A-B**) Cohesion loss monitored in mid-M phase cells. The number of GFP spots was scored in G1 arrested cells and mid-M phase cells. The percentage of cells with 2 GFP spots was plotted. 100–200 cells were scored for each data point and data was generated from two independent experiments. (**A**) Scc2p-AID or Scc3p-AID depletion induces cohesion loss in strains with normal or fusion cohesin. Top panel is strains with normal cohesin and bottom panel is fusion cohesin strains. WT (White), *SCC2-AID* (black) and *SCC3-AID* (gray). (**B**) Pds5p-AID depletion induces cohesion loss in strains with normal or fusion cohesin. Top panel is strains with normal cohesin and bottom panel is fusion cohesin strains. WT (White) and *PDS5-AID* (black). (**C-D**) mid-M phase arrested cells fixed and processed for ChIP using anti-Mcd1p antibodies as described in Materials and methods. Mcd1p binding was assessed by qPCR and presented as percentage of total DNA using the same primer pairs at each site. Left Panel is chromosome III peri-centric region (*CARC1*), middle panel is chromosome IV arm CAR region (*TRM1*) and right panel is regions immediately adjacent to *CEN4* and *CEN14*. (**C**) Scc2p-AID or Scc3p-AID depletion induces loss of fusion cohesin and normal cohesin binding to DNA. Top panel is strains with normal cohesin and bottom panel is fusion cohesin strains. WT (gray), *SCC2-AID* (black) and *SCC3-AID* (red). (**D**) Pds5p-AID depletion reduces the amount of fusion cohesin and normal cohesin binding to DNA. Top panel is strains with normal cohesin and bottom panel is fusion cohesin strains. WT (gray), *PDS5-AID* (black) and *SMC3-AID* (red).

DOI: https://doi.org/10.7554/eLife.46347.004

The following figure supplements are available for figure 3:

**Figure supplement 1.** Characterizing the effect of Scc2p-AID or Scc3p-AID depletion in normal and fusion cohesin strains.

DOI: https://doi.org/10.7554/eLife.46347.005

**Figure supplement 2.** Characterizing the effects of Pds5p-AID depletion in normal and fusion cohesin strains.

DOI: https://doi.org/10.7554/eLife.46347.006

and arm CAR *TRM1* and dramatically reduced binding near *CEN4* and *CEN14* (**Figure 3C**). The lack of cohesin binding to DNA was not due to loss of either Mcd1p or Smc3p Mcd1p fusion as shown by western blot analysis (**Figure 3—figure supplement 1E**). Thus, Scc2p and Scc3p are essential for both normal and fusion cohesin to bind chromosomes. The robust fusion cohesin binding to DNA means the Smc3p Mcd1p interface is not the sole an entry gate for DNA. The fact that Scc2p and Scc3p are required for fusion cohesin binding to DNA means they do not function solely to prevent DNA escape through the putative Smc3p Mcd1p 'exit' gate.

We also assessed the effect of Pds5p-AID depletion on cohesin binding in mid-M arrested cells bearing normal cohesin or with fusion cohesin. Pds5p-AID depletion was confirmed by both western blot and loss of Pds5p binding to chromosomes as assayed by ChIP (**Figure 3—figure supplement 2B,D**). There was a decrease in Mcd1p of normal cohesin and a more modest decrease of fusion protein (**Figure 3—figure supplement 2C**). This result is consistent with Pds5p serving partially to protecting Mcd1p from factors that degrade it (**Stead et al., 2003**; **D'Ambrosio and Lavoie, 2014**). At the pericentric *CARC1*, Pds5p-AID depletion reduced normal and fusion cohesin by 60–75%, but the reduction for normal and fusion cohesin was more severe at *TRM1* (**Figure 3D**). Adjacent to *CENs*, normal cohesin binding was unperturbed whereas fusion cohesin binding was reduced ~50%. Thus, the fusion did not suppress the decrease in cohesin binding to DNA caused by Pds5p-AID depletion. These results show that Pds5p promotes cohesion and cohesin binding to DNA by a mechanism other than preventing DNA exit through the putative Smc3p Mcd1p gate.

## Smc3 head acetylation in the fusion is required for efficient generation of sister chromatid cohesion but not for cell viability

The exit gate model posited that Wpl1p opens the Smc3p Mcd1p interface to allow DNA escape from cohesin, but gate opening is blocked by Eco1p mediated acetylation of Smc3p at K112 K113 (**Chan et al., 2012**; **Beckouët et al., 2016**). Loss of Eco1p function is lethal and causes defects in cohesion and condensation in budding yeast (**Skibbens et al., 1999**; **Tóth et al., 1999**; **Bloom et al., 2018**). Previous work showed that the fusion suppressed the lethality and condensation defects of *eco1Δ* (**Chan et al., 2012**; **Bloom et al., 2018**). However, the *eco1Δ* fusion strain had an increased defect in sister chromatid cohesion compared to the wild-type fusion (**Bloom et al., 2018**). Eco1p acetylates Smc3p at multiple sites (**Unal et al., 2008**). We wanted to see whether the partial suppressive effects of the fusion were due solely to bypassing the requirement for acetylation at the essential K112 K113 residues.

We addressed whether the fusion suppresses the need for K112 K113 acetylation by placing the *smc3 K112R K113R* (RR) mutant in the fusion and assessing the effect on cohesin function. If Smc3p K112 K113 acetylation functions solely to keep this interface closed, then the fusion should

completely suppress the inviability and cohesion defects caused by the RR mutations. Strains bearing the RR fusion as the sole Smc3p and Mcd1p source are viable but are sensitive to cold and microtubule destabilizing drug benomyl (*Figure 4A and B*). Thus, the fusion suppressed the inviability that occurs when Smc3p head acetylation is blocked, just as it suppressed inviability of an *eco1Δ*, but defects in cohesin function remained. Therefore, Smc3p K112 K113 acetylation has biological functions in addition to controlling the dissociation of the Smc3p Mcd1p interface.

We also examined sister chromatid cohesion and cohesin binding to chromosomes in fusion wild-type and RR strains. Cells were synchronously arrested in mid-M phase as described for AID

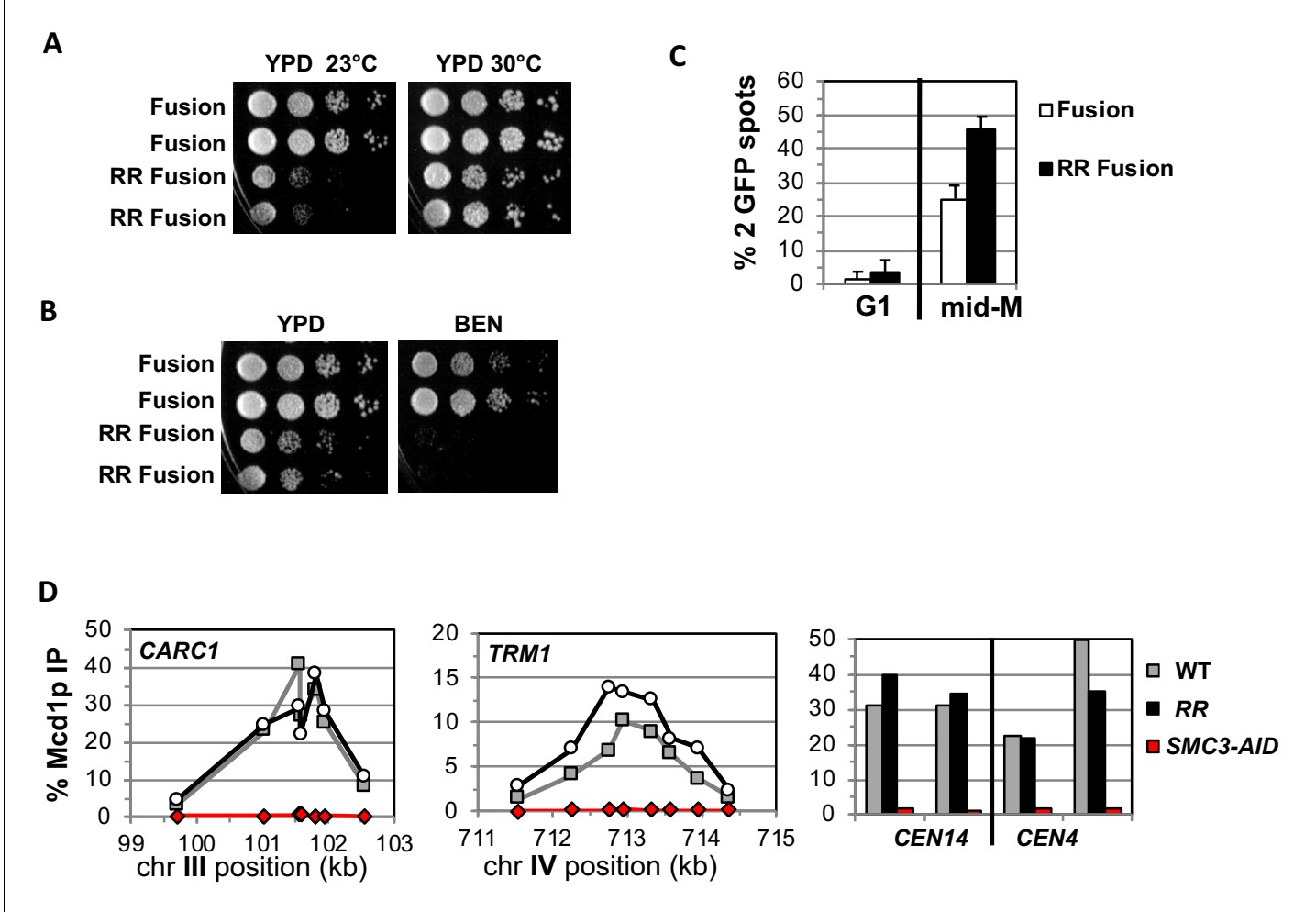

**Figure 4.** Fusion cohesin with the smc3 K112R K113R mutations are viable but have defects in growth defects in both growth and sister chromatid cohesion. (A–B) Fusion cohesion bearing *smc3-K112R, K113R* (RR) mutations are viable but cold sensitive and benomyl sensitive. Haploid strains with wild-type fusion cohesin (VG3940-2D) or RR fusion (VG3930-5C) were grown to saturation at 30°C then plated at 10-fold serial dilution onto (A) YPD at either 23°C or 30°C and incubated for 3 days. (B) YPD alone or YPD containing benomyl at 12.5 μg/ml (BEN) then incubated for 23°C for 4 days. (C–D) Strains in A-B were synchronously arrested in mid-M as described in *Figure 3* above except cells were grown at 30°C and auxin was omitted. Cells were fixed and processed to assess cohesion and for ChIP. (C) The RR mutation in fusion cohesin causes an increased defect in cohesion fusion. Cohesin was scored and plotted as described in *Figure 3*. Fusion cohesin (White) and RR fusion cohesin (black). (D) Fusion and RR fusion cohesin bind DNA at similar levels. mid-M phase arrested cells were fixed and processed for ChIP using anti-Mcd1p antibodies as described in Materials and methods and *Figure 3*. Left Panel is chromosome III peri-centric region (*CARC1*), middle panel is chromosome IV arm CAR region (*TRM1*) and right panel is regions immediately adjacent to *CEN4* and *CEN14*. WT fusion (gray), RR fusion (black) and *SMC3-AID* (red).

DOI: https://doi.org/10.7554/eLife.46347.007

The following figure supplement is available for figure 4:

**Figure supplement 1.** Characterization of fusion wild-type and *smc3-K112R K113R* (RR) mutations.

DOI: https://doi.org/10.7554/eLife.46347.008

depletion experiments above except auxin was omitted and cells grown at 30°C due to the severe growth defect of RR fusion strains at 23°C. Forty-five percent of the smc3p-RR fusion bearing cells had a cohesion defect, an ~2 fold increase compared to wild-type fusion cells (*Figure 4C*). This increase suggests that Smc3 head acetylation regulates cohesion by a mechanism other than or in addition to blocking DNA exit through the putative gate. The cohesion defect could also explain the benomyl sensitivity as microtubule destabilization would abrogate early S phase bipolar attachments necessary for proper segregation in cells with severe cohesion defects (*Guacci and Koshland, 2012*). Finally, wild-type fusion and smc3p-RR fusion cohesin bound to chromosomes at similar levels and these fusion proteins are also present in cells at similar levels (*Figure 4D*, *Figure 4—figure supplement 1A–B*). This similarity means the increased cohesion defects in the RR fusion was not due to decreased cohesin binding to DNA. These results indicate that Smc3p head acetylation promotes efficient sister chromatid cohesin independent of stabilizing DNA binding by keeping Smc3p Mcd1p interface closed.

## Smc3p Mcd1p interface controls the integrity of another cohesin interface to allow cohesin binding to DNA

Having established that key cohesin regulators function independently of preventing DNA exit through the Smc3p Mcd1p interface, we turned to characterize the function of the interface itself. Mutations in Mcd1p or Smc3p that affect residues in the interface are lethal (*Arumugam et al., 2006*; *Eng et al., 2014*; *Gligoris et al., 2014*; *Robison et al., 2018*). One such mutation, *smc3-L1029R* was thought to prevent cohesin binding to DNA because when tagged with GFP, it failed to immunolocalize to the cohesin-rich centromere cluster (*Gligoris et al., 2014*). This mutation also abolished a chemical crosslink normally detected between Smc3p and Mcd1p (*Gligoris et al., 2014*). These results were interpreted to mean that interface mutants eliminated Smc3p Mcd1p association, thereby allowing DNA to escape from cohesin entrapment. However, our results with fusion cohesin show that Pds5p and Scc3p promote DNA binding by mechanisms independent keeping this interface 'gate' closed. Their binding proximal to the Smc3p Mcd1p interface raised the alternative possibility that this interface has a gate-independent biochemical property that regulates cohesin binding to DNA.

To test this possibility, we examined mutations that alter amino acids of the interface from both Mcd1p (*mcd1-L75K* and *mcd1-L89K*) and Smc3p (*smc3-I1026R* and *smc3-L1029R*). All four mutations were previously shown to be unable to support viability (*Arumugam et al., 2006*; *Gligoris et al., 2014*). Three possible outcomes are expected from the fusion mutants. If all these mutations abrogate cohesin function solely by preventing closure of a putative Smc3p Mcd1p interface exit gate, then all four mutants should be suppressed by the fusion. Alternatively, if the fusion fails to suppress all these mutants, then these residues must modulate cohesin in a way distinct from keeping a putative exit gate closed. Finally, the fusion could suppress mutants in only Smc3p or only Mcd1p. For example, if Mcd1p residues only promote association with Smc3p, whereas Smc3p residues have a second function, then the fusion would suppress only the *mcd1* mutations.

Before testing these mutants in the fusion, we further characterized them in normal cohesin to provide a baseline for judging their effects on fusion cohesin. The *smc3* mutants were placed into a haploid strain bearing *SMC3-3V5-AID* (*SMC3-AID*) as the sole *SMC3* whereas the *mcd1* mutants were placed in a haploid bearing *MCD1-AID* as the sole *MCD1*. The wild-type *SMC3* allele or *MCD1* allele was placed into the *SMC3-AID* or *MCD1-AID* strain for positive controls, respectively. The parent *SMC3-AID* or *MCD1-AID* strain alone served as the negative controls. Strains were tested by dilution plating on auxin containing media to assess viability under *AID* depletion conditions. As expected, *SMC3-AID* alone or *MCD1-AID* alone were inviable on auxin, whereas those also containing wild-type alleles are viable (*Figure 5A and B* and (*Eng et al., 2014*). None of the mutant alleles rescued the inviability of the *AID* strains, corroborating previous studies that these interface residues provide an essential cohesin function (*Figure 5A and B*).

We next examined whether the four interface mutants compromised sister chromatid cohesion. Cells were synchronously arrested in mid-M phase under AID depletion conditions and arrest confirmed by FACS (Materials and methods; *Figure 5—figure supplement 1A–B*). We assessed cohesion at the *LYS4* locus using the LacO-GFP, LacI system. As expected, almost all *SMC3-AID* and *MCD1-AID* cells had 2 GFP spots indicating a total loss of cohesion, whereas cells with the respective wild-type alleles promote cohesion so few cells had 2 GFP (*Figure 5C and D*). All four mutant

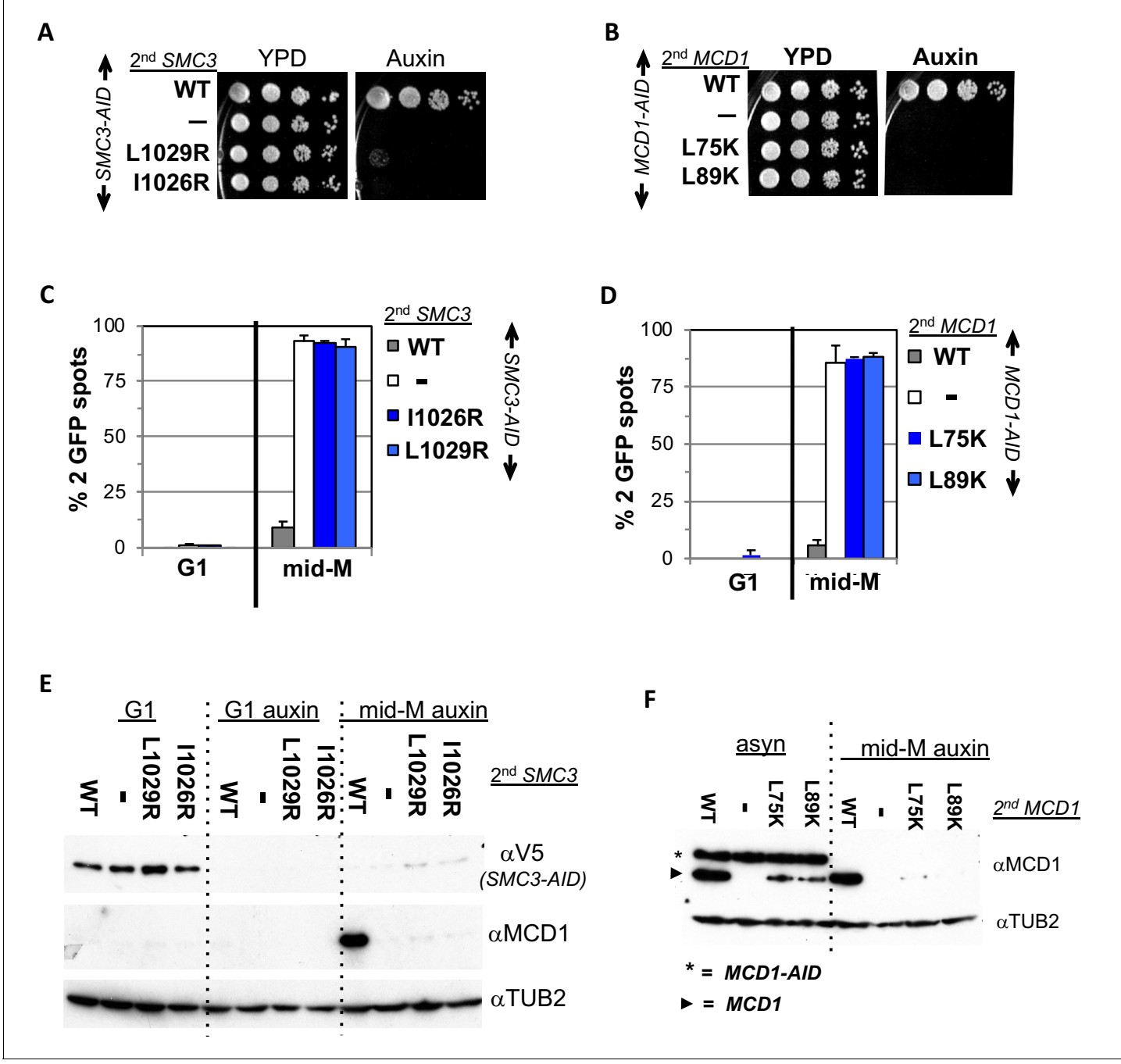

**Figure 5.** Smc3p coiled-coil and Mcd1p NHD interface residues are required for cohesin function and integrity. (A) Viability loss of Smc3p interface mutant strains. Haploid *SMC3-AID* strain alone (VG3651-3D) or containing either wild-type (WT; BRY474), *smc3-I1026R* (VG3905-7A) or *smc3-L1029R* (BRY492) were grown and plated as described in **Figure 2**. (B) Viability loss of Mcd1p interface mutant strains. Haploid *MCD1-AID* strain alone (VG3902-3A) or containing, either wild-type (WT; VG3914-2C), *mcd1-L75K* (VG3916-5B) or *mcd1-L89K* (VG3918-9D) were grown and plated as described in **Figure 2**. (C–D) Haploids in A and B were synchronously arrested in mid-M phase as described in **Figure 3**. The number of GFP spots was scored in G1 arrested cells and mid-M phase cells. The percentage of cells with 2 GFP spots was plotted. 100–200 cells were scored for each data point and data was generated from two independent experiments. (C) Cohesion loss in *smc3* interface mutant mid-M cells. *SMC3-AID* alone (white), or containing WT (gray), *smc3-I1026R* (dark blue) and *smc3-L1029R* (light blue). (D) Cohesion loss in *mcd1* mutant mid-M cells. *MCD1-AID* strain alone (white) or containing, either WT (gray), *mcd1-L75K* (dark blue) or *mcd1-L89K* (light blue). (E–F) Protein extracts from the synchronous arrest regimen in A and B were made from G1 cells, auxin treated G1 cells and mid-M cells then subjected to western blot analysis. (E) Mcd1p is degraded in mid-M phase *smc3* interface mutants. Top panel detects Smc3p-3V5-AID (αV5), middle panel detects Mcd1p (αMcd1). Tubulin (αTub2p; bottom panel) was used as a loading control. As a control we showed in (**Figure 6—figure supplement 2**) that the *smc3-L1029R* mutation did not affect its protein levels by

*Figure 5 continued on next page*

*Figure 5 continued*

comparing 6 HA-epitope-tagged Smc3p (VG3943-1C) and Smc3p-L1029R (VG3944-3D). (**F**) Mcd1p interface mutants are degraded in mid-M cells. Top panel detects Mcd1p (αMcd1) with star indicating Mcd1p-AID and arrow head Mcd1p WT and interface mutants, bottom panel detects Tubulin (αTub2p). .

DOI: https://doi.org/10.7554/eLife.46347.009

The following figure supplement is available for figure 5:

**Figure supplement 1.** FACS analysis showing cell cycle arrest of normal cohesin strains bearing interface mutations.

DOI: https://doi.org/10.7554/eLife.46347.010

alleles were totally defective in cohesion, similar to what was seen in the *AID* alone strains (*Figure 5C and D*).

To investigate why this cohesin defect was so severe, we examined whether these mutants support the integrity of the cohesin trimer. Mcd1p is not detectable from anaphase through G1 due to degradation but is present from the S phase through mid-M. However, Mcd1p presence from S phase through mid M is dependent upon its binding to both Smc1p and Smc3p as Mcd1p is rapidly degraded if either Smc1p-AID or Smc3p-AID are depleted (*Çamdere et al., 2015*; *Guacci et al., 2015*). Therefore, Mcd1p presence serves as an in vivo readout for presence of cohesin trimer (Smc3p, Mcd1p, Smc1p).

We used this readout to assess the impact of *smc3* and *mcd1* interface mutants on trimer formation. Mcd1p was present In mid-M cells when cells had Smc3p (*SMC3 SMC3-AID*) but absent when Smc3p was missing (*SMC3-AID*) as expected (*Figure 5E*). Mcd1p was absent in cells expressing only smc3p-L1029R or smc3p-I1026R (*Figure 5E*). Similarly, levels mcd1p-L75K and mcd1p-L89K were greatly reduced (*mcd1-L75K MCD1-AID* and *mcd1-L89K MCD1*-AID) (*Figure 5F*, right side). Our results show that perturbation of the Smc3p Mcd1p interface with mutations in either Smc3p or Mcd1p led to Mcd1p degradation.

The Mcd1p degradation explained the loss of all cohesin function in the *smc3* and *mcd1* mutants but also complicated interpretations of the function of the interface. These mutations could indeed abrogate stable association of Smc3p and Mcd1p leading to Mcd1p degradation, or simply alter the interface in a way that allowed protease accessibility to the Mcd1p N terminus leading its degradation. In either case, the loss of Mcd1p in the normal strains precluded determining whether the interface performs another function distinct from an exit gate.

With this foundation, we examined the properties of fusions containing either *smc3-I1026R* or *smc3-L1029R* in an *SMC3-AID* background, and either *mcd1-L75K* or *mcd1-L89K* mutations in an *MCD1-AID* background. These strains allowed us to examine the properties of the mutated fusions in the absence of the corresponding wild-type normal (unfused) subunit after auxin addition. Dilution plating onto auxin containing media revealed that the fusion failed to suppress the inviability of any of the *smc3* or *mcd1* interface mutants (*Figures 6A* and *7A*). We then tested whether the fusion suppressed the degradation of Mcd1p in *smc3* and *mcd1* interface mutants. All four mutant fusion proteins were present in mid-M cells, although the mcd1p fusion mutants were slightly less abundant than the wild-type fusion (*Figure 6B and 7B*). Cell cycle arrest was confirmed by FACS (*Figure 6— figure supplement 1*, *Figure 7—figure supplement 1*). The presence of the mutant fusion proteins demonstrated that the complete Mcd1p loss observed when normal cohesin contains interface mutations depended upon a free N terminus of Mcd1p. More importantly, the inviability of the fusion mutants was not due to absence of fusion protein, but rather due to some other biochemical defect caused by the mutations.

We then examined the four mutant fusions for sister chromatid cohesion and cohesin binding to chromosomes in cells synchronously arrested in mid-M under auxin depletion conditions. All four mutant fusion strains had almost no sister chromatid cohesion when depleted for Mcd1p-AID or Smc3p-AID (*Figures 6C* and *7C*). By ChIP, all four fusion mutants exhibited complete loss of cohesin binding to *CARs* and *CEN* DNA when depleted for Mcd1p-AID or Smc3p-AID (*Figures 6D* and *7D*). The loss of cohesin binding to DNA explains why the interface fusion mutants were inviable and unable to generate cohesion.

The shared dramatic phenotypes of all four mutant fusions lead to several important conclusions about the Smc3p Mcd1p interface. First, the amino acids of Mcd1p and Smc3p that make up the interface must have essential biochemical activities other than directly maintaining the topological

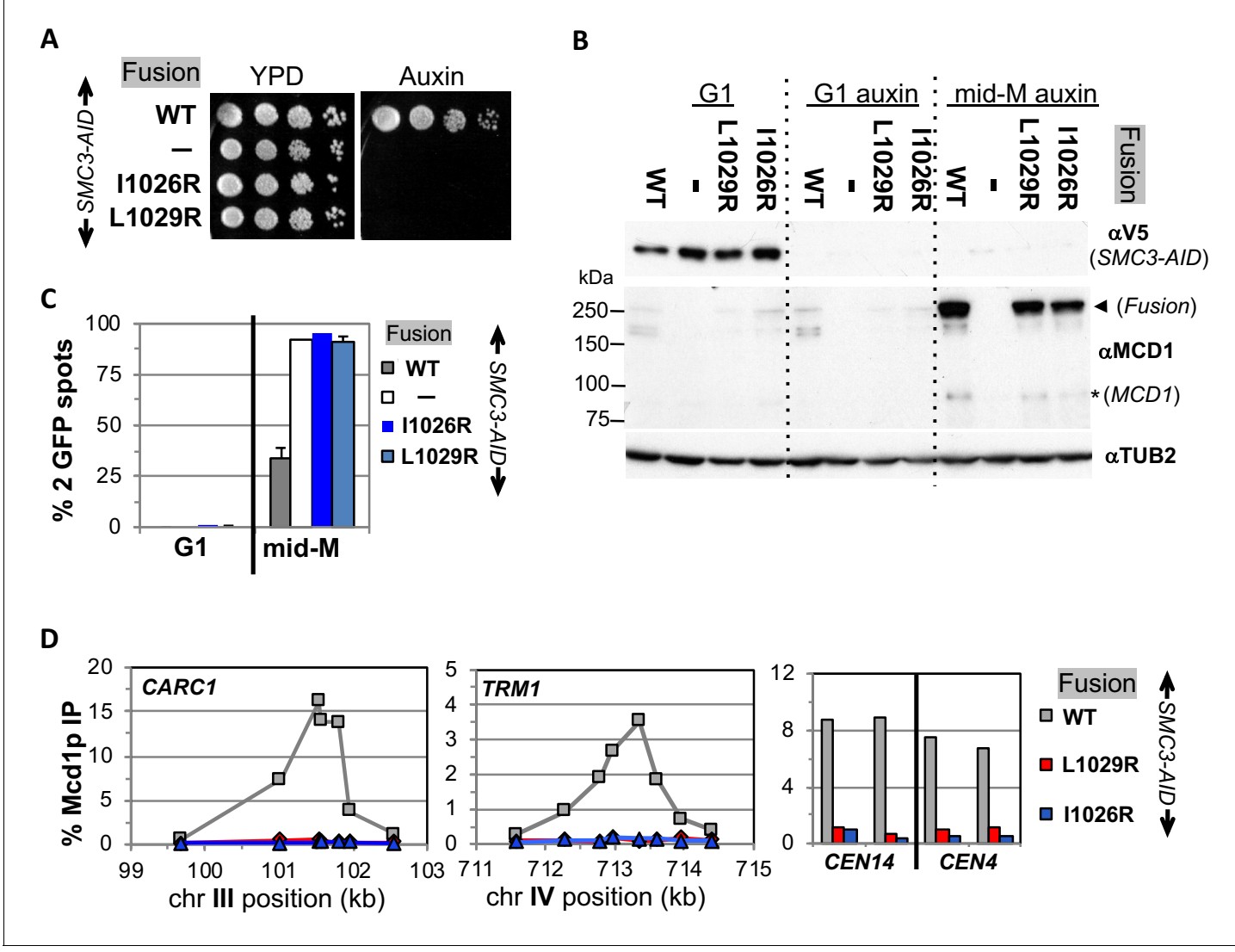

**Figure 6.** Fusion cohesin requires Smc3p coiled coil residues for function. (A) Viability loss of Smc3p coiled coil mutant fusion strains. Haploid *SMC3-AID* strain alone (VG3651-3D) or containing fusion cohesin, either wild-type (WT; VG3694-7C), *smc3-I1026R* (VG3908-17B) or *smc3-L1029R* (VG3872-3B) were grown and plated as described in *Figure 2*. (B–D) Haploids in A were synchronously arrested in mid-M phase as described in *Figure 3*. (B) Fusion WT and *smc3* interface mutant proteins are present in mid-M cells. Protein extracts from G1 and auxin treated G1 and mid-M phase arrested cells were subjected to western blot analysis using antibodies against V5 to detect Smc3p-3V5-AID (αV5; top panel) and against Mcd1p (αMcd1p; middle panel). Arrow head indicates fusion protein and star (*) indicates normal Mcd1p. Tubulin (αTub2p; bottom panel) was used as a loading control. (C) Cohesion is completely lost in *smc3* mutant fusion mid-M phase arrested cells. The number of GFP spots was scored in G1 arrested cells and mid-M phase cells. The percentage of cells with 2 GFP spots was plotted. *SMC3-AID* alone (white), or containing fusion WT (gray), *smc3-I1026R* (dark blue) and *smc3-L1029R* (light blue). 100–200 cells were scored for each data point and data was generated from two independent experiments. (D) fusion *smc3* interface mutant cohesin fails to bind DNA in mid-M phase. Fusion cohesin binding to chromosomes was assayed by ChIP using anti-Mcd1p antibodies as described in *Figure 3*. WT fusion (gray), *smc3-I1026R* fusion (blue) and *smc3-L1029R* fusion (red).

DOI: https://doi.org/10.7554/eLife.46347.011

The following figure supplements are available for figure 6:

**Figure supplement 1.** Confirming cell cycle arrest of fusion cohesin bearing smc3 interface mutants.
DOI: https://doi.org/10.7554/eLife.46347.012

**Figure supplement 2.** Western blot to assess the effect of smc3-L1029R interface mutant on cohesin levels.
DOI: https://doi.org/10.7554/eLife.46347.013

**Figure supplement 3.** Fusion wild-type and fusion L1029 mutant coimmunoprecipitate with Scc2p and Scc3p equally well.
DOI: https://doi.org/10.7554/eLife.46347.014

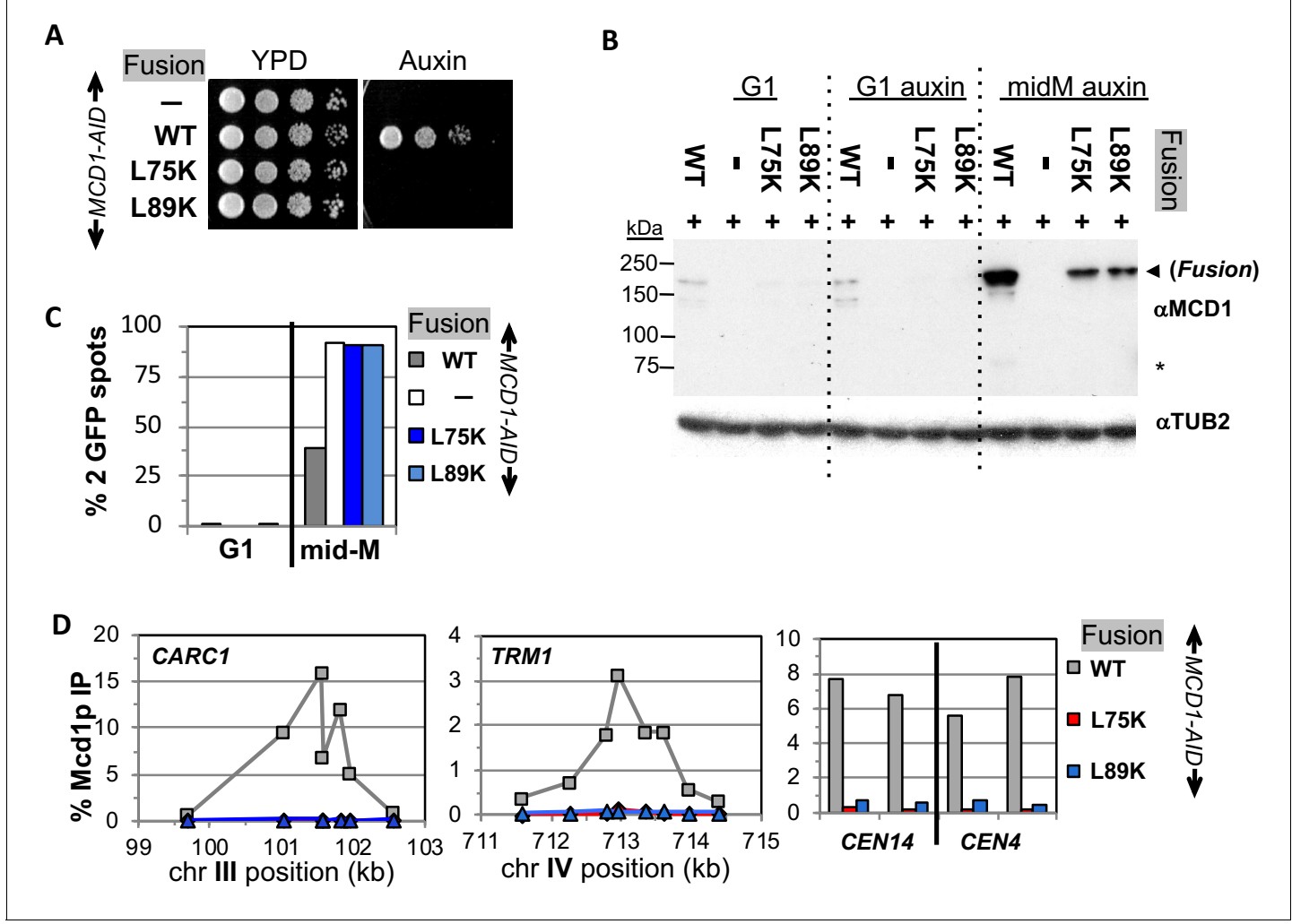

**Figure 7.** Fusion cohesin requires Mcd1p NHD residues for function. (A) Viability loss of Mcd1p NHD mutant fusion strains. Haploid *MCD1-AID* strain alone (VG3902-3A) or containing fusion cohesin, either wild-type (WT; VG3937-2C), *mcd1-L75K* (VG3938-3A) or *mcd1-L89K* (VG3939-7B) were grown and plated as described in *Figure 2*. (B–D) Haploids in (A) were synchronously arrested in mid-M phase as described in *Figure 3*. (B) Fusion WT and mcd1 NHD mutant proteins are present in mid-M cells. Protein extracts from G1, auxin-treated G1 and mid-M phase cells were subjected to western blot analysis using antibodies against Mcd1p (αMcd1p; top panel). Arrow head indicates fusion protein and star (*) indicates normal Mcd1p. Tubulin (αTub2p; bottom panel) was used as a loading control. (C) Cohesion is completely lost in *mcd1* mutant fusion mid-M cells. The number of GFP spots was scored in G1 arrested cells and mid-M phase cells. The percentage of cells with 2 GFP spots was plotted. Haploid *MCD1-AID* strain alone (white) or containing fusion cohesin, either WT (gray), *mcd1-L75K* (dark blue) or *mcd1-L89K* (light blue). 100–200 cells were scored for each data point. (D) fusion *mcd1 NHD* mutant cohesin fails to bind DNA in mid-M phase. Fusion cohesin binding to chromosomes was assayed by ChIP using anti-Mcd1p antibodies as described in *Figure 3*. WT (gray), *mcd1-L75K* (red) and *mcd1-L89K* (blue).

DOI: https://doi.org/10.7554/eLife.46347.015

The following figure supplement is available for figure 7:

**Figure supplement 1.** Confirming cell cycle arrest of fusion cohesin bearing mcd1 NHD interface mutants.

DOI: https://doi.org/10.7554/eLife.46347.016

integrity of the cohesin ring at this interface. Second, the fact that mutations in Smc3p Mcd1p interface residues have the same phenotypes suggests that these Smc3p and Mcd1p regions collaborate to perform the same biological function, likely through their interaction. Finally, the absence of DNA binding in the mutant fusions suggests that at least one essential biochemical activity of the interface is to promote DNA entrapment by modulating the opening or stable closing of a distal interface of cohesin. This modulation of a distal interface could be achieved directly through changing cohesin structure or indirectly through a regulator.

## Discussion

Elucidating how cohesin carries out its many biological activities requires understanding how cohesin's topological entrapment of DNA is established, maintained and released. A reversible dissociation of the interface between Smc3p and Mcd1p has been proposed to be either an entrance gate or an exit gate enabling cohesin to establish or maintain topological DNA entrapment, respectively (*Chan et al., 2012*; *Murayama and Uhlmann, 2015*; *Beckouët et al., 2016*). Here, we assessed gate-independent functions of this interface in yeast using a fusion protein where Smc3p at its C-terminus was fused to Mcd1p at its N-terminus. Our results reveal surprising phenotypes that provide significant new insights into the mechanism and regulation of topological entrapment.

We show that fusion cohesin binds DNA with the same distribution and at similar if not higher levels as wild-type cohesin. Furthermore, we corroborate previous finding that fusion cohesin is capable of robust sister chromatid cohesion (*Gruber et al., 2006*; *Bloom et al., 2018*). These two features coupled with the requirement for topological entrapment to generate cohesion, strongly suggests that DNA efficiently enters cohesin through an interface other than the Smc3p Mcd1p interface to become topologically entrapped.

We cannot rule out that the Smc3p Mcd1p interface could act as an alternative secondary entrance gate for DNA. However, recent observations show SMC complexes are DNA translocases (*Merkenschlager and Nora, 2016*; *Terakawa et al., 2017*; *Wang et al., 2017*). This complicated activity likely requires a spatially constrained sequence of reaction intermediates that would be incompatible with promiscuous mechanisms for DNA binding and entrapment.

Surprisingly, we also find that fusion cohesin containing mutations in the Smc3p coil or Mcd1p NHD interface completely fails to bind DNA as assayed by ChIP. Therefore, this interface is essential for cohesin to DNA binding even though itself is not a DNA entrance gate. A complete absence of ChIP signal is also seen when cohesin DNA entrapment is abrogated by depleting cohesin subunits or cohesin loader (This study and *Eng et al., 2014*; *Çamdere et al., 2015*). In contrast, *eco1* mutants destabilize cohesin binding to DNA yet exhibit cohesin ChIP signals equivalent to wild-type, likely because transient cohesin entrapment is preserved by formaldehyde crosslinking (*Lengronne et al., 2006*; *Noble et al., 2006*). Therefore, the severity of the phenotype of mutations in the Smc3p Mcd1p interface suggest that this interface normally promotes the establishment of DNA entrapment. Since DNA cannot enter through the Smc3p and Mcd1p in fusion cohesin, this interface must promote the establishment of cohesin topological entrapment by allowing DNA to enter cohesin through either Smc1p Mcd1p or hinge dimer interface. We favor the hinge dimer interface because sealing this interface in vivo causes inviability and inhibits cohesin DNA binding, whereas fusing the Smc1p Mcd1p interface had no effect on viability (*Gruber et al., 2006*).

The mechanism by which the Smc3p Mcd1p interface promotes the opening or closing of a distal interface of the cohesin remains to be determined. The interface could promote DNA binding indirectly by recruiting factors like Scc2p or Scc3p, which are essential for cohesin loading onto DNA. However, fusion cohesin and the *smc3-L1029R* fusion mutant bind Scc2p and Scc3p equally well (*Figure 6—figure supplement 3A–C*), disfavoring this model.

Alternatively, we prefer a model in which the Smc3p-Mcd1p interface directly modulates a change in cohesin conformation that promotes topological entrapment. Different conformations of cohesin and other SMC complexes have been observed and to be dependent upon the head ATPase (*Onn et al., 2008*). Based upon these observations, we suggest that the ATPase alters Smc1p interaction with the C terminus of Mcd1p which in turn alters the coiled coil through the interaction of the Mcd1-NHD with the Smc3p coiled coil. The subsequent change in the coiled coil alters a distal interface to stimulate DNA binding (*Figure 8A*). By this model, Scc2p and Scc3p could alter the interface through their ability to stimulate the ATPase activity of cohesin (*Murayama and Uhlmann, 2014*), or through their binding to Mcd1p.

Our analysis of the fusion cohesin in cells defective for Pds5p and Smc3p acetylation also suggest novel functions for the interface in post-DNA binding functions of cohesin. Previously, these factors were suggested to stabilize DNA entrapment by inhibiting exit of DNA through the interface between Smc3p and Mcd1p. This model predicted that the fusion of Smc3p and Mcd1p should suppress Pds5p or acetylation defects. However, we demonstrate here that the Smc3p-Mcd1p fusion fails to suppress any phenotypes of Pds5p depletion, and only partially suppresses the phenotypes

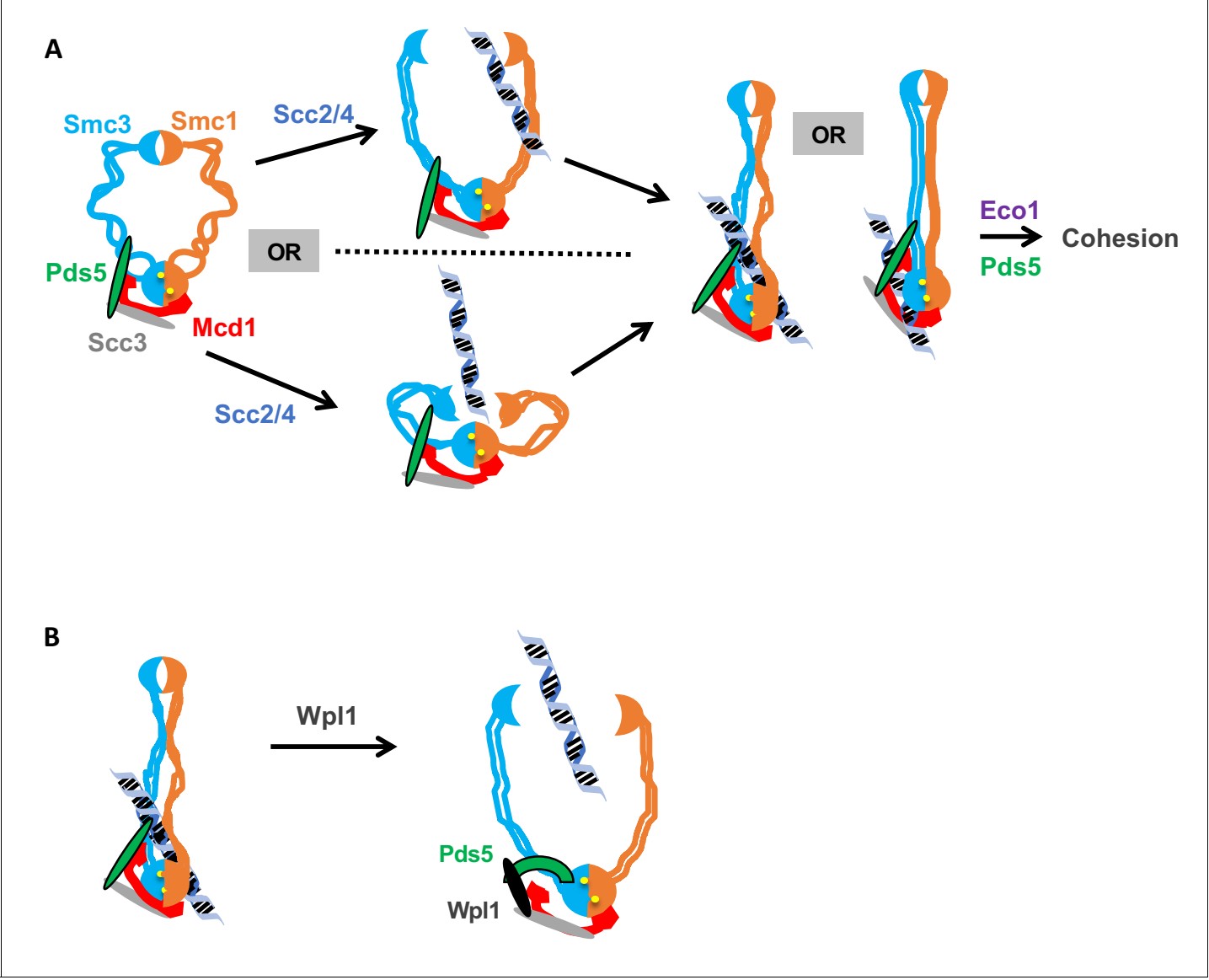

**Figure 8.** Model for how the Smc3p Mcd1p interface regulates the hinge dimer interface to control cohesin DNA binding. (**A**) *Left side*: cohesin complex in an open ring conformation. *Middle top*: cohesin loader Scc2p/Scc4p binding at the Smc3p Mcd1p interface (not shown) triggers a conformation change that opens the hinge dimer (half-moons) interface, which allows DNA (blue-black helix) to enters cohesin. Alternatively, this conformational change could generate a folded ring to bring the hinges into close proximity to Scc2/Scc4 bound near the head, thereby enabling it to directly act on the hinges. *Right side*: DNA interaction with cohesin and Pds5p (green oval) binding at Smc3p Mcd1p interface triggers another conformational change that traps DNA either near the head (left) or in the small lumen formed by Mcd1p binding both the Smc1p head and the Smc3p coiled-coil (right). (**B**) *Left side*: Cohesin stably bound to DNA. *Right side*: Wpl1p (black oval) binds Pds5p (green arc) and cohesin at its Smc3p Mcd1p interface and triggers a conformation change that opens hinge dimer (half-moons) interface and allows DNA to escape. Pds5p has both positive and negative functions so it is depicted differently with the green oval showing its positive role in promoting DNA binding and the green arc showing its negative role as it binds Wpl1p. We note that recent in vitro evidence suggests DNA may first interact with cohesin at the Scc3p and Mcd1p region entry into the trimer. This earlier step is not depicted and has no bearing on our model.

DOI: https://doi.org/10.7554/eLife.46347.017

of Smc3p lacking K112, K113 acetylation. Thus, these regulators have post-DNA binding functions independent of preventing DNA exit through the Smc3p Mcd1p interface.

While these post-DNA binding functions of Pds5p and Smc3p acetylation could be totally independent of the Smc3p Mcd1p interface, a more parsimonious model would be that these regulators act through the interface by modulating its ability to alter cohesin conformation. Precedence for the

Smc3p Mcd1p interface playing a role in post DNA binding comes from the study of DNA damage induced cohesion in mid-M. In undamaged cells, cohesin loads onto DNA in mid-M but fails to generate cohesion (*Ström et al., 2007*; *Unal et al., 2007*). However, upon DNA damage, post-translational modifications of the Mcd1-NHD make cohesin that is loaded in mid-M phase become cohesive (*Heidinger-Pauli et al., 2009*). Thus, modification of the Smc3p Mcd1p interface can modulate cohesin function after DNA binding in response to DNA damage. This result raises the possibility that the Smc3p Mcd1p interface also modulates post-DNA binding functions of cohesin in general.

The ability of the Smc3p Mcd1p interface to alter a distal interface required for DNA binding also suggests a potential alternative mechanism of Wpl1p-induced removal of cohesin from DNA. The existing model is based upon the observation that Wpl1p has been shown to destabilize the binding of an N-terminal Mcd1p fragment to cohesin (*Beckouët et al., 2016*). This observation has been interpreted as strong evidence that Wpl1p opens the Smc3p Mcd1p interface to allow DNA to escape topological entrapment (*Beckouët et al., 2016*). Our data do not exclude this 'exit gate' model. However, it is worth noting that structural and crosslinking studies suggests that Pds5p binds both to Mcd1p near the NHD and to Smc3p proximal to interaction with the Mcd1p NHD (*Chan et al., 2013*; *Eng et al., 2014*); Huis in 't *Huis in 't Veld et al., 2014*). Consequently, Pds5p provides a second link between Mcd1p and Smc3p distinct from the Smc3p Mcd1p interface. Because of this second link, destabilization of the Smc3p Mcd1p interface would not be sufficient to allow DNA escape through this interface. As an alternative model, we suggest Wpl1p-mediated dissociation of the Smc3p Mcd1p interface abrogates DNA entrapment by promoting DNA escape through the same distal interface (likely the hinge) by which DNA enters to establish entrapment (*Figure 8B*). If so, the phenotype of mutations that compromise the stability of the hinge dimer could be suppressed by removing dimer antagonists like Wpl1p. Indeed, in *S. pombe*, the inviability of mutations in the Smc1p or Smc3p hinge can be suppressed by deleting *WPL1* (*Xu et al., 2018*; *Xu and Yanagida, 2019*). Therefore, the Smc3p Mcd1p interface is not itself an exit gate but rather a Wpl1p-dependent mediator of a distal exit gate.

In summary, all Smc complexes have a conserved interface analogous to the interface between the N terminus of Mcd1p and the coiled coil of Smc3p (*Gligoris and Löwe, 2016*). Bacterial Smc complex and the yeast condensin can bind and then rapidly translocate along DNA (*Merkenschlager and Nora, 2016*; *Terakawa et al., 2017*). Toggling these two activities will likely require conformational changes in the coiled coils. Given the conservation of Smc interfaces, these conformational changes are likely to be controlled by regulators acting on the interface analogous to the Smc3p Mcd1p interface.

# Materials and methods

## Key resources table

| Reagent type or resource | Designation | Source or reference | Identifiers | Additional information |
|---|---|---|---|---|
| Genetic reagent (*S. cerevisiae*) | NCBITaxon:4932 | this paper | Yeast strains | *Supplementary file 1* |
| qPCR primers | DNA primer | IDT | | *Supplementary file 2* |
| Antibody | Rabbit polyclonal Anti-Mcd1p | V. Guacci (via Covance) | RbαMcd1p (555) | WB (1:10,000) ChIP (1:1,000) |
| Antibody | Rabbit polyclonal Anti-Pds5p | V. Guacci (via Covance) | RbαPds5p (556) | WB (1:20,000) ChIP (1:1,000) |
| Antibody | Mouse monoclonal Anti-MYC | Roche | MαMYC (9E10) Cat#116666006001 | WB (1:10,000) IP (1:667) |
| Antibody | Mouse monoclonal Anti-HA | Roche | MαHA (12CA5) Cat#11667203001 | WB (1:10,000) |
| Antibody | Mouse monoclonal Anti-V5 | Invitrogen | MαV5 Cat# 46–0705 | WB (1:10,000) |

*Continued on next page*

*Continued*

| Reagent type or resource | Designation | Source or reference | Identifiers | Additional information |
|---|---|---|---|---|
| Antibody | Goat polyclonal HRP Anti-rabbit | Biorad | Cat# 170–6515 | WB (1:10,000) |
| Antibody | Goat polyclonal HRP Anti-mouse | Biorad | Cat# 170–6516 | WB (1:10,000) |
| Antibody | Rabbit polyclonal Anti-Tub2p | P. Meluh (via Covance) | RbαTUB2 | WB (1:40,000) |
| Dynabeads | Protein A | Invitrogen | Ref# 10002D | IP: Use 50 ul/IP |
| Dynabeads | Protein G | Invitrogen | Ref# 10004D | IP: Use 50 ul/IP |
| Chemical compound | Auxin (3-indole acetic acid) | Sigma | Cat# C9911 | 750 µM for plates 500 µM for liquid |
| Chemical compound | Alpha factor | Sigma | αF (αFactor) Cat# T6901 | |
| Chemical compound | Nocodazole | Sigma | Nz Cat# M1404 | |

## Yeast strains and media

Yeast strains used in this study are A364A background, and their genotypes are listed in *Supplementary file 1*. SC minimal and YPD media were prepared as described (*Guacci et al., 1997*). Auxin (3-indole acetic acid; Sigma-Aldrich Catalog# I3705) Benomyl (a gift from Dupont) and camptothecin (Sigma catalog# C9911) plates used to assess drug sensitivity were prepared as previously described (*Guacci and Koshland, 2012*). Preparation of auxin containing media for depletion of AID-tagged proteins was as previously described (*Eng et al., 2014*).

## Dilution plating assays

Cells were grown to saturation in YPD media at 23˚C (or 30˚C when listed) then plated in 10-fold serial dilutions. Cells were incubated on plates at relevant temperatures or containing drugs as described.

## Synchronous arrest in mid-M phase under auxin depletion conditions
### G1 arrest

Asynchronous cultures of cells were grown to mid-log phase at 23˚C in YPD media, then alpha factor (αFactor) (Sigma; Catalog# T6901) was added to 10–8M. Cells were incubated for 23˚C for 3 hr (or at 30˚C for 2.5 hr) to induce arrest in G1 phase. For depletion of AID-tagged proteins, auxin was added (500 µM final) and cells incubated an additional 1 hr in αFactor containing media.

### Synchronous arrest in mid-M phase

G1 arrested cells were washed 3x in YPD containing 0.1 mg/ml Pronase E (Sigma; Catalog# P6911), once in YPD, then resuspended in YPD containing nocodozale (Sigma; Catalog# M1404) at 15 µg/ml final. For depletion of AID-tagged proteins, auxin was added (500 µM final) in all wash media and in resuspension media containing nocodazole to ensure depletion at all times. Cells were incubated at 23˚C for 3 hr (or 30˚C for 2.5 hr) to arrest in mid-M phase.

## Protein extracts and western blotting
### Total protein extracts

Cell equivalents of 2 to 4 $OD_{600}$ were washed in cold 1xPBS, pelleted in a microfuge at 10 k for 1 min then quick frozen using liquid nitrogen and stored at −80˚C. Cell extracts were made as described in *Eng et al. (2015)* with the following modifications. Initial lysis in an eppendorf tube was in 200 µl 20% Trichloroacetic acid (W/V) then 500 µl 5% TCA added twice, and all liquid combined in a new eppendorf and treated as described. The final protein pellet was resuspended in 212 µl 2x Laemmli buffer +26 µl 1M Tris buffer pH8.

## Immunoprecipitation

Cell equivalents of 20 $OD_{600}$ were washed in cold 1xPBS, pelleted in a microfuge at 10 k for 1 min then quick frozen using liquid nitrogen and stored at −80℃. Cells were lysed and cleared extracts incubated with Mouse anti-MYC (9E10) antibodies (Roche) to immunoprecipitate MYC epitope tagged proteins as previously described (*Bloom et al., 2018*). Total extracts for these experiments were taken from cleared lysates before addition of antibodies.

## Western blots

Protein extracts were loaded onto 8% SDS page gels, subjected to electrophoresis then transferred to PDVF membranes using standard laboratory techniques.

## Monitoring cohesion using LacO-GFP assay

Cohesion was monitored using the LacO-LacI system. Briefly, cells contained a GFP-LacI fusion and tandem LacO repeats were integrated at *LYS4*, located 470 kb from *CEN4.* Cells were fixed and processed to allow the number of GFP signals in each cell to be scored and the percentage of cells with 2-GFP spots determined as previously described (*Guacci and Koshland, 2012*).

Chromatin Immunoprecipitation (ChIP) was performed as previously described (*Robison et al., 2018*). Primers used for ChIP are shown in *Supplementary file 2*.

## Microscopy

Images were acquired with a Zeiss Axioplan2 microscope (100X objective, NA = 1.40) equipped with a Quantix CCD camera (Photometrics).

Flow cytometry analysis was performed as previously described (*Bloom et al., 2018*).

## Plasmid constructs

Site directed mutagenesis using the Stratagene Quick-change kit was employed to generate the *smc3-I1026R* and *smc3-L1029R* alleles in normal cohesin on *LEU2* integrating plasmids pVG419 or in fusion cohesin on *LEU2* or *URA3* integrating plasmids pVG511 or p4897, respectively. Similarly, we generated *mcd1-L75K* and *mcd1-L75K* alleles in normal cohesin on *LEU2* integrating plasmids pJH18 or in fusion cohesin on *LEU2* integrating plasmids pVG511, respectively. The *smc3-K112R, K113R* mutant in fusion cohesin on *URA3* integrating plasmid p4897. The *smc3* and *mcd1* mutations were confirmed by sequencing the entire ORF and the promoter region to ensure it was the only change.

## Strain construction

### AID-tagged proteins

Details about the Auxin-mediated destruction of AID-tagged proteins in yeast was previously described (*Eng et al., 2014*). Briefly, the *TIR1* E3-ubiquiting ligase placed under control of the GPD promoter and marked by *C. glibrata TRP1* replaced the *TRP1* gene on chromosome IV. Strains bearing genomic copies AID tagged of alleles of *SCC2*, *SCC3* and *PDS5* as sole source were made by C-terminal tag with *3V5-AID2* sequences using standard PCR techniques into TIR1 bearing yeast to generate *SCC2-AID*, *SCC3-AID* and *PDS5-AID* strains, respectively. *MCD1-AID* alleles were built the same way except using AID1 and did not have the 3V5 tag (*Eng et al., 2014*). The *SMC3-AID*-tagged strains were built as previously described (*Guacci et al., 2015*).

# Acknowledgements

We thank Rebecca Lamothe, Lorenzo Constantino and Siheng Xiang for their critical reading of the manuscript. This work was funded by National Institutes of Health Grant (1R35 GM-118189-01 to DK).

## Additional information

### Funding

| Funder | Grant reference number | Author |
|---|---|---|
| National Institutes of Health | 1R35 GM-118189-01 | Douglas E Koshland |

The funders had no role in study design, data collection and interpretation, or the decision to submit the work for publication.

### Author contributions

Vincent Guacci, Conceptualization, Data curation, Formal analysis, Supervision, Validation, Investigation, Methodology, Writing—original draft, Project administration, Writing—review and editing; Fiona Chatterjee, Data curation, Formal analysis, Methodology, Writing—review and editing; Brett Robison, Data curation, Formal analysis, Investigation, Methodology, Writing—review and editing; Douglas E Koshland, Conceptualization, Supervision, Funding acquisition, Investigation, Methodology, Writing—original draft, Project administration, Writing—review and editing

### Author ORCIDs

Vincent Guacci ⓘ https://orcid.org/0000-0003-0281-713X
Douglas E Koshland ⓘ https://orcid.org/0000-0003-3742-6294

### Decision letter and Author response

Decision letter https://doi.org/10.7554/eLife.46347.022
Author response https://doi.org/10.7554/eLife.46347.023

## Additional files

### Supplementary files

• Supplementary file 1. Yeast strains.
DOI: https://doi.org/10.7554/eLife.46347.018

• Supplementary file 2. Primers used for chromatin immunoprecipitation (ChIP).
DOI: https://doi.org/10.7554/eLife.46347.019

• Transparent reporting form
DOI: https://doi.org/10.7554/eLife.46347.020

### Data availability

All data generated in this study are included in the manuscript.

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
