## [Decision Letter]

Thank you for submitting your article "Communication between distinct subunit interfaces of the cohesin complex promotes its topological entrapment of DNA" for consideration by *eLife*. Your article has been reviewed by three peer reviewers, and the evaluation has been overseen by Prasad Jallepalli as the Reviewing Editor and Jessica Tyler as the Senior Editor. The following individuals involved in review of your submission have agreed to reveal their identity: Kerry Bloom (Reviewer #1); Susannah Rankin (Reviewer #3).

The reviewers have discussed the reviews with one another and the Reviewing Editor has drafted this decision to help you prepare a revised submission.

Summary:

This is an excellent manuscript that tests several key assumptions in the prevailing models about cohesin function. The authors employ a Smc3-Mcd1 fusion to covalently bond the putative DNA gate and proceed to test several models using various mutant combinations that are proposed to regulate the opening and closing of the DNA gate. The findings suggest that the Smc3-Mcd1 interface has important functions that are distinct from gating, and they demonstrate that DNA can enter the cohesin ring even when this interface cannot be opened. Overall the study is well executed and important for the field. The reviewers advise the following essential revisions.

Essential revisions:

1) Even though the chromosome localization of the fusion protein is largely normal, the loading dynamics can be very different from the WT. Because this fusion protein cannot be unloaded by Wapl, it has a longer residence time on chromosomes. Defective/slower loading of the fusion may be masked by its inability to be released. Without kinetic data, the authors should interpret their results more cautiously. It remains possible that the Smc3-Scc1N interface is a DNA entry gate. If the gate is sealed by the fusion, an alternative gate is used, but the fusion is loaded less efficiently and with slower kinetics. This possibility needs to be discussed, and the conclusion stated with greater precision (the Smc3-Scc1N interface cannot be the sole, functional entry gate).

2) While the requirement for the intact Smc3-Scc1N interface is well demonstrated, the mechanism by which this interface exerts its function is not explored. The Discussion and model in Figure 8A are too speculative. This needs to be toned down.

3) The speculation in Figure 8B contradicts data from multiple labs. The fusion protein is more stable on chromosomes based on FRAP data in yeast. The N-terminal fragment of Scc1 is dissociated from Smc3 in a Wapl-dependent manner both in vitro and in yeast cells. Pds5 binding and Smc3 binding to Scc1N are mutually exclusive. These and other data show that the Smc3-Scc1N interface is the DNA exit gate. It should be removed or at least explained against the backdrop of the aforementioned and conflicting evidence.

4) The authors should comment on whether the 20-aa linker is expected to reconstitute the Smc3-Mcd1 interface, given its length versus the distance between the corresponding residues in structural models of the native interface.

5) It would be helpful if the authors commented on whether or not their findings support the alternative "hold and release" model described in Xu et al., 2018.

---

## [Author Response]

Essential revisions:1) Even though the chromosome localization of the fusion protein is largely normal, the loading dynamics can be very different from the WT. Because this fusion protein cannot be unloaded by Wapl, it has a longer residence time on chromosomes. Defective/slower loading of the fusion may be masked by its inability to be released. Without kinetic data, the authors should interpret their results more cautiously. It remains possible that the Smc3-Scc1N interface is a DNA entry gate. If the gate is sealed by the fusion, an alternative gate is used, but the fusion is loaded less efficiently and with slower kinetics. This possibility needs to be discussed, and the conclusion stated with greater precision (the Smc3-Scc1N interface cannot be the sole, functional entry gate).

The revised Discussion brings up the possibility of an alternative entry gate but we do explain why we don’t favor that idea.

2) While the requirement for the intact Smc3-Scc1N interface is well demonstrated, the mechanism by which this interface exerts its function is not explored. The Discussion and model in Figure 8A are too speculative. This needs to be toned down.

The model is indeed speculative but it does fit the data. We deemphasize it in the revised Discussion by not describing the model in detail but merely referring to the model figure. The purpose of the model is in part to emphasize the reviewer’s point that the interface could function by multiple mechanism that hopefully will stimulate the reader to test.

3) The speculation in Figure 8B contradicts data from multiple labs. The fusion protein is more stable on chromosomes based on FRAP data in yeast. The N-terminal fragment of Scc1 is dissociated from Smc3 in a Wapl-dependent manner both in vitro and in yeast cells. Pds5 binding and Smc3 binding to Scc1N are mutually exclusive. These and other data show that the Smc3-Scc1N interface is the DNA exit gate. It should be removed or at least explained against the backdrop of the aforementioned and conflicting evidence.

None of the data from other labs proves that the Smc3p-Mcd1p interface is the exit gate. We have tried to clarify this point in the Discussion. What has been shown is that destabilization of this interface causes cohesin to dissociate from DNA. These results mean either the interface is the exit gate or a regulator of different interface that is the “real” exit gate. We provide one caveat to the previous interpretation that it is the exit gate. Pds5p by binding to Mcd1p and Smc3p provides a backup to the interface to prevent DNA from exiting even if the interface dissociates. The reviewer cites an observation that suggests that Pds5p binds to the Mcd1p NHD. The crosslinking between Pds5p and Smc3p is not at the interface but rather further along the coiled-coil, therefore this binding would not be mutually exclusive. Second this model has never made sense with the fact that Pds5p promotes DNA binding and cohesion maintenance in vivo as we have shown previously and show again here.

Another point we do not mention in the manuscript is that the stability of yeast fusion cohesin by FRAP is also consistent with our model. In our model, dissociation of the interface leads to opening of another interface. The interface is more stable in the fusion because Mcd1p and Smc3p are held in close proximity, so they reassociate very quickly.

4) The authors should comment on whether the 20-aa linker is expected to reconstitute the Smc3-Mcd1 interface, given its length versus the distance between the corresponding residues in structural models of the native interface.

The spacer between Smc3p and Mcd1p is 132bp encoding 44 amino acids (linker + 3TEV sites). It would be worth discussing if the fusion failed to function, but obviously that is not the case. Gruber et al., 2006, made a fusion with an even longer spacer and that still functions. It might be interesting to shorten the spacer to see what happens, but it is irrelevant to the paper and discussion here.

5) It would be helpful if the authors commented on whether or not their findings support the alternative "hold and release" model described in Xu et al., 2018.

Thanks for pointing out that omission. We have added comments about the suppressors that Dr. Yanadiga’s lab that support our view that DNA could escape through the hinge dimer.